# Learning Depth-regularized Radiance Fields from Asynchronous RGB-D Sequences

## Abstract

Recently it is shown that learning radiance fields with depth rendering and depth supervision can effectively promote the view synthesis quality and convergence. But this paradigm requires input RGB-D sequences to be synchronized, hindering its usage in the UAV city modeling scenario. To this end, we propose to jointly learn large-scale depth-regularized radiance fields and calibrate the mismatch between RGB-D frames. Although this joint learning problem can be simply addressed by adding new variables, we exploit the prior that RGB-D frames are actually sampled from the same physical trajectory. As such, we propose a novel **time-pose function**, which is an implicit network that maps timestamps to SE(3) elements. Our algorithm is designed in an alternative way consisting of three steps: (1) time-pose function fitting; (2) radiance field bootstrapping; (3) joint pose error compensation and radiance field refinement. In order to systematically evaluate under this new problem setting, we propose a large synthetic dataset with diverse controlled mismatch and ground truth. Through extensive experiments, we demonstrate that our method outperforms strong baselines. We also show qualitatively improved results on a real-world asynchronous RGB-D sequence captured by drones. Codes, data, and models will be made publicly available.

## 1 Introduction

Incorporating depth rendering and depth supervision into radiance fields has been demonstrated as a helpful regularization technique in several recent studies [2, 21, 32, 20]. However, this technique has not yet been successfully introduced into radiance field learning from UAV (Unmanned Aerial Vehicle) images, despite it's a typical choice in city modeling. A closer look at the aforementioned works reveals that they assume synchronized RGB and depth signals, which is hard to guarantee in UAV vision due to the lack of suitable synchronized sensors for long sensing ranges. So we study the *problem* of learning depth-regularized radiance fields from asynchronous RGB-D sequences.

As a recap, the canonical radiance field [15] learns a neural network parameterized by $\theta$ that represents a 3D scene, from input images $I$ and their intrinsic/extrinsic parameters $\mathcal{T}_I$. To alleviate the reliance on $\mathcal{T}_I$, some works [30, 11, 8] aim to resolve a different problem that self-calibrates $\mathcal{T}_I$. In other words, they jointly learn $\theta$ and $\mathcal{T}_I$ from input images $I$. Similarly, the *formulation* considered here is to learn scene representation $\theta$, camera parameters $\mathcal{T}_I$ and $\mathcal{T}_D$ from inputs images $I$ and depths $D$.

It is natural to develop the joint learning *formulation* to resolve the *problem*, as it amounts to adding new parameters to existing methods [30, 11, 8]. However, an important prior is ignored that RGB-D frames are actually sampled from the same physical trajectory. As conceptually shown in Fig. 1-a/b, $\mathcal{T}_I$ and $\mathcal{T}_D$ can be considered as samples from a function that maps timestamps to SE(3) elements. We name this function as **time-pose function** and model it with a neural network parameterized by $\phi$. As such, we address the *problem* with a *new formulation* that learns scene representation $\theta$ and

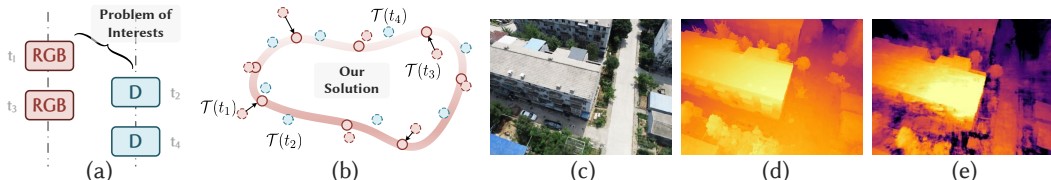

Figure 1: The problem of interest is to learn a depth-regularized radiance field using asynchronous RGB-D sequences (a). We proposed a time-pose function as conceptually shown in (b) to leverage the prior that RGB-D seuqnces are actually sampled from the same physical underlying trajectory. For a novel view (c), our method can render a better depth map (d) than Mega-NeRF (e).

time-pose function $\phi$ from inputs RGB images $I$ and depths $D$. An interesting fact is that both $\theta$ and $\phi$ are implicit neural representation networks (or say coordinate-based networks) that allow fully differentiable training. To our knowledge, this *new formulation* has not been proposed before.

We also propose an effective learning scheme designed in an alternative manner. In the first stage, we fit the time-pose function $\phi$ using one modality (e.g., RGB images) and infer the poses of the other, using a balanced pose regression loss and a speed regularization term. Secondly, we bootstrap a large-scale radiance field $\theta$ based upon Mega-NeRF [29] using the outputs of the trained time-pose function. Thanks to the first step, depth regularization can be imposed here in spite of RGB-D misalignment. Finally, thanks to the cascade of two fully differentiable implicit representation networks, we jointly optimize the 3D scene representation $\theta$ and compensate pose errors by updating $\phi$.

Since the *problem* considered is new, we contribute a synthetic dataset (named AUS) for systematic evaluation. Using six large-scale 3D scenes, realistic drone trajectories of different difficulty levels are generated. Specifically speaking, simple trajectories are heuristically designed with a zig-zag pattern while complicated ones are generated by manual control signals in simulation. We also control the mismatch between RGB-D sequences using different protocols, to cover as many as possible scenarios that the algorithm may encounter in reality. Through a set of comprehensive experiments, we show the proposed method outperforms several state-of-the-art counterparts and our design choices contribute positively to performance. Last but not least, we present a real-world evaluation using asynchronous sensors on drones. Our depth rendering results (on unseen viewpoint) is shown in Fig. 1-d, which is much better than the result of Mega-NeRF shown in Fig. 1-e. This success is credited to the usage of depth regularization as made possible by our novel algorithm.

To summarize, we have the following contributions in this paper: (1) We formalize the new *problem* of learning depth-regularized radiance fields from asynchronous RGB-D sequences, which is rooted in UAV city modeling. (2) We identify an important domain-specific prior in this problem: RGB-D frames are sampled from the same underlying trajectory. We instantiate this prior into a novel time-pose function and develop a cascaded fully differentiable implicit representation network. (3) In order to systematically study the problem, we contribute a photo-realistically rendered synthetic dataset that simulates different types of mismatch. (4) Through a comprehensive benchmarking on this new dataset and real-world asynchronous RGB-D sequences, we demonstrate that our method can promote performance over strong prior arts. Anonymous code: https://anonymous.4open.science/r/async-nerf

## 2   Related Works

**Large-scale Radiance Fields.** Neural Radiance Field (NeRF) [15] has shown impressive results in neural reconstruction and rendering. However, its capacity to model large-scale unbounded 3D scenes is limited. Several strategies [29, 26, 32, 14, 35] have been proposed to address this limitation, with a common principle of dividing large scenes into blocks or decomposing the scene into multiple levels. Block-NeRF [26] clusters images by dividing the whole scene according to street blocks. Mega-NeRF [29] utilizes a clustering algorithm that partitions sampled 3D points into different NeRF submodules. BungeeNeRF [32] trains NeRFs using a growing model of residual blocks with predefined multiple scales of data. Switch-NeRF [14] designs a gating network to jointly learn the scene decomposition and NeRFs without any priors of 3D scene shape or geometric distribution. However, these prior works fail to leverage the rich geometric information in depth images for effective regularization.

**Depth-regularized Radiance Fields.** Volumetric rendering requires extensive samples and sufficient views to effectively differentiate between empty space and opaque surfaces. Depth maps can serve as geometric cues, providing regularization constraints and sampling prior, which accelerates NeRF's convergence towards the correct geometry. DS-NeRF [3] enhances this process using depth supervision from 3D point clouds, estimated by structure-from-motion, and a specific loss for rendered ray termination distribution. Mono-SDF [38] and Dense-Depth Prior [21] further supplement this with a pretrained dense monocular depth estimator for less-observed and textureless areas. To adapt NeRF for outdoor scenarios, URF [20] rasterizes a pre-built LiDAR point cloud map to generate dense depth images and alleviates floating elements by penalizing floaters in the free space. Moreover, S-NeRF [34] completes depth on sparse LiDAR point clouds using a confidence map, effectively handling street-view scenes with limited perspectives. However, those methods are not readily applicable to UAV captured images due to the lack of suitable synchronized sensors for long ranges.

**Broader UAV Vision and Synchronization.** Like autonomous driving, UAV vision is drawing increasing attention due to its unique characteristics. Broader UAV vision covers many topics like counting [31][7], trajectory forecasting [18], intention prediction [33], object tracking [16], physics understanding [39], next-best-view prediction [6], 3D reconstruction [41], and calibration [19]. Sensor synchronization is challenging for UAV vision (and other settings) and several works address the problem from an algorithmic perspective. One possibility is to adopt tailored hardware designs or software protocols [1] to synchronize all the devices. Another branch of sensor-agnostic methods utilizes temporal priors by using Sum-of-Gaussians [4] or parametric interpolation functions [36].

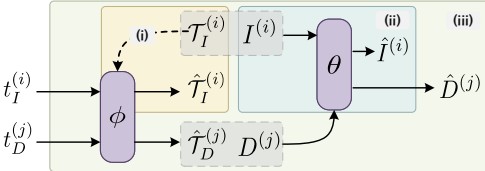

Figure 2: **Three-step Optimization.** (i) A time-pose function parameterized by $\phi$ is trained to predict camera poses from timestamps; (ii) The neural radiance field parameterized by $\theta$ is bootstrapped with pure RGB losses; (iii) Both of the parameters $\theta, \phi$ are jointly optimized with RGB-D supervision.

## 3  Problem Formulation and Optimization Pipline

**Problem & Challenge.** Our goal is to learn a neural radiance field parameterized by $\theta$ for large-scale scene representation from UAV images as done in prior works [29, 32]. However, these prior works fail to leverage depth supervision, which is known [3, 21] as useful for training floater-less NeRFs. To our knowledge, there are no easily accessible synchronized RGB-D sensor suites for **large-scale** outdoor scenes, and trivially synchronizing them according to timestamp[1] cannot fully address the misalignment issue. Instead of using expensive hardware, we take an algorithmic perspective.

**Input & output.** There are some prior works on large-scale scene modeling using aerial images [32, 29, 5, 6]. In this study, we assume an input RGB-D stream captured by drones: a set of RGB camera images $\{I^{(i)}\}_{i=1}^{N_I}$ and a set of depth maps $\{D^{(j)}\}_{j=1}^{N_D}$ (shown in Fig. 1-a) and we aim to recover the spatiotemporal transformations between them. Given that our focus is on relative transformation, it is viable to consider either the RGB or the depth stream as the reference without compromising generality. For convenience, we assume a set of camera poses $\{\mathcal{T}_I^{(i)}\}_{i=1}^{N_I}$ for color images are obtained by an SfM algorithm. The neural scene representation parameterized by $\theta$ outputs an image $\hat{I}$ as well as a depth map $\hat{D}$ at a given perspective camera pose $\mathcal{T}$.

**Observation.** Note that all the sensor data are captured with a drone on the **same** trajectory[2], we can model the relationship between capture time $t$ and sensor poses $\mathcal{T}$ with an implicit **time-pose function** as $\phi : t \to \hat{\mathcal{T}} = [\hat{\mathbf{x}}, \hat{\mathbf{q}}]$, where $t$ is the timestamp of capture, and the estimated pose $\hat{\mathcal{T}}$ is represented by a translation vector $\hat{\mathbf{x}} \in \mathbb{R}^3$ and a quaternion $\hat{\mathbf{q}} \in \mathbb{R}^4$.

**Pipeline overview.** We formulate our method as a 3-step optimization problem (as shown in Fig. 2). First, since the time-pose relationship for the RGB captures are given, we can train a time-pose

---

[1]The so-called soft synchronization.

[2]but they are not necessarily synchronized in terms of acquisition time.

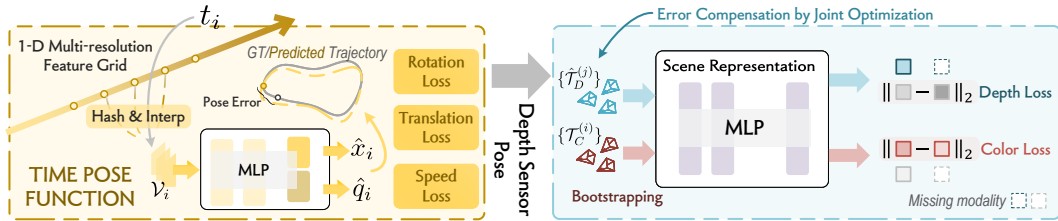

Figure 3: **Method Pipeline.** The time-pose function is modeled using a 1-D multi-resolution hash grid with direct and speed losses. After bootstrapping the scene representation networks with pure RGB signals, the predicted depth sensor poses are used for jointly optimizing the NeRFs' parameters $\theta$. At each timestamp ($t_i$ from RGB sequence or $t_j$ from depth sequence), only one modality of sensor signals is provided, thus only one loss term is activated (shown on the right).

function on the RGB sequence (Fig. 2-(i)). Then, to train the neural radiance field, we first bootstrap the network with pure RGB supervision (Fig. 2-(ii)). To further enable training with RGB-D supervision, we can use the previously trained time-pose function to estimate the corresponding depth camera poses $\{\mathcal{T}_D^{(j)}\}$ of the depth timestamps $\{t_D^{(j)}\}$. Since both of the networks are differentiable, we jointly optimize the networks in an end-to-end manner in the third stage (Fig. 2-(iii)).

## 4 Method

We introduce in Section 4.1 the details of learning an implicit time-pose function. In Section 4.2, we describe our neural scene representation networks and the bootstrapping strategy. In Section 4.3, we adopt depth supervision and jointly train the time-pose function with RGB-D pairs.

### 4.1 Time-Pose Function

We represent the camera trajectory as an implicit time-pose function $\phi$ whose input is a timestamp $t$, and whose output is a 6-DoF pose $\mathcal{T}$ that consists of a 3-D translation $x_i$ and a 4-D quaternion $q_i$.

**Network Overview** The time-pose function (shown in the left part of Fig. 3) is approximated with a compact 1-D multi-resolution hash grid $\{\mathcal{G}^{(l)}\}_{l=1}^{L}$, followed by an MLP decoder. The hash grid consists of $L$ levels of separate feature grids with trainable hash encodings [17]. The reason why we choose this architecture is as follows: The time-pose function is a coordinate-based function that may contain coarse and fine-level feature components [3][27], and this architecture allows us to sample the hash encodings from each grid layer with different resolutions and perform quadratic interpolation on the extracted encodings to obtain a feature vector $\mathcal{V}_i$ when querying a specific timestamp $t$ that is in the range of all timestamps. This design choice is also empirically validated in Table. 3.

After obtaining the interpolated feature vector, an MLP with two separated decoder heads is used to predict the output translation $\hat{x}_i$ and rotation $\hat{q}_i$ vectors respectively. The forward pass can be expressed in the following equations:

$$\mathcal{V}_i = \mathcal{F}_{\text{MLP}} \left( \text{concat}\{\text{interp}(\text{h}(t; \pi_l), \ \mathcal{G}_\theta^l\}_{l=1}^{L}; \Phi_{\text{MLP}} \right), \tag{1}$$

$$\hat{\mathcal{T}}_i = [\hat{x}_i, \hat{q}_i] = l_{\text{trans}}(\mathcal{V}_i; \Phi_{\text{trans}}), \ l_{\text{rot}}(\mathcal{V}_i; \Phi_{\text{rot}}), \tag{2}$$

where interp denotes interpolation, h is the hash function parameterized by $\pi_l$, $\mathcal{F}_{\text{MLP}}, l_{\text{trans}}, l_{\text{rot}}$ are the MLP networks and the decoder heads, with $\Phi_{\text{MLP}}, \Phi_{\text{trans}}, \Phi_{\text{rot}}$ representing their parameters.

**Depth-pose Prediction** Since both the depth maps and the RGB images are collected by the same drone on the same flight, they cover the same spatial-temporal footprints except for the difference in the placement of the two sensors on the aircraft. For every depth frame, we first predict the RGB camera pose using the capture timestamps of the depth sensor with the time-pose function then transform the predicted RGB camera pose to the depth sensor pose with a pre-calibrated pose transformation $\mathcal{T}_{RGB \rightarrow D}$ between sensors.

**To optimize the Time-Pose Function,** we propose the following objective function:

$$\mathcal{L} = \lambda_{\text{trans}}\mathcal{L}_{\text{trans}} + \lambda_{\text{rot}}\mathcal{L}_{\text{rot}} + \lambda_{\text{speed}}\mathcal{L}_{\text{speed}}, \tag{3}$$

---

[3]This is shown by the ground truth trajectory in Fig. 4 and the predicted trajectory in the supplementary.

where $\mathcal{L}_{\text{trans}}, \mathcal{L}_{\text{rot}}, \mathcal{L}_{\text{speed}}$ are translation, rotation and speed losses respectively as shown in the left panel of Fig. 3. and $\lambda_{\text{trans}}, \lambda_{\text{rot}}, \lambda_{\text{speed}}$ are the weighting parameters. Note that $\lambda_{\text{trans}}$ and $\lambda_{\text{rot}}$ are automatically adjusted as explained in a later paragraph.

**Pose Representation.** There are some common choices to represent rotation for optimizing camera poses like rotation matrices [37] or Euler-angles [25, 28]. However, they are not continuous for representing rotation [42] due to their non-homeomorphic representation space to $\mathrm{SO}(3)$. We choose to use unit quaternion as our raw rotation representation because arbitrary 4-D vectors can be easily mapped to legitimate rotations by normalizing them to the unit length [10].

**Optimization of Translation and Rotation.** We optimize the translation and the rotation vectors by minimizing the mean square error (MSE) between the estimated and ground-truth camera poses:

$$\mathcal{L}_{\text{trans}} = \frac{1}{n} \sum_{i=1}^{n} (x_i - \hat{x}_i)^2, \ \mathcal{L}_{\text{rot}} = \frac{1}{n} \sum_{i=1}^{n} (q_i - \hat{q}_i)^2. \tag{4}$$

Since $x$ and $q$ are in different units, the scaling factor $\lambda_{\text{trans}}$ and $\lambda_{\text{rot}}$ play an important role in balancing the losses. To prevent translation and rotation from negatively influencing each other in training and to tap into possible mutual facilitation, we make the weighting factors learnable by using homoscedastic uncertainty [9] as $\mathcal{L}_{\sigma} = \mathcal{L}_{\text{trans}} \exp(-\hat{s}_{\text{trans}}) + \hat{s}_{\text{trans}} + \mathcal{L}_{\text{rot}} \exp(-\hat{s}_{\text{rot}}) + \hat{s}_{\text{rot}}$, where $\hat{s}$ are learnable parameters, thus the loss terms are balanced during training course[4].

**Optimization of Motion Speed.** Observing that the time-pose function is essentially a function of translational displacement and angular displacement with respect to time, we can use the average linear velocity[5] to supervise the gradient of the network output, with regard to the input vectors. Since the linear velocity variation is small and the angular velocity variation is relatively larger in the scenes captured by the drone, only the average linear velocity is used to supervise the neural network and the latter is not supervised in our method:

$$\mathcal{L}_{\text{speed}} = \mathrm{MSE}(v(t_i), \hat{v}(t_i)) = \frac{1}{n} \sum_{i=1}^{n} (v(t_i) - \frac{\partial \hat{x}}{\partial t}(t_i))^2, \ v(t_i) = \frac{\partial x}{\partial t} \Big|_{t=t_i} \approx \frac{x_i - x_{i-1}}{t_i - t_{i-1}} \tag{5}$$

### 4.2 Bootstrapping Large-scale Neural Radiance Fields

In this part, we introduce our proposed scene representation (right half of Fig. 3) that is bootstrapped in the second phase of the optimization process (Fig. 2-(ii)). Due to the limited capacity of MLPs, we follow Mega-NeRF[29] and partition the scene map into a series of equal-sized blocks in terms of spatial scope, and each block learns its individual scene representation with an implicit field. In this stage, we optimize the scene representation with pure RGB data. Specifically, the radiance field is denoted as $\{f_{\text{NeRF}}^{(i)}\}_{i=1}^{N_x \times N_y}$, where $N_x, N_y$ denotes the spatial grid size. Each implicit function represents a geographic region with $\mathbf{x}_i^{\text{centroid}}$ as its centroid. The $k$th scene model can be written as:

$$f_{\text{NeRF}}^{(k)}(\gamma(\mathbf{x}_{\text{pts}}), \gamma(\mathbf{d})) \to (\hat{c}, \sigma), \tag{6}$$

where $\mathrm{k} = \arg\min_{j} ||\mathbf{x}_{\text{pts}} - \mathbf{x}_j^{\text{centroid}}||_2$ and $\gamma$ is the positional encoding function.

For view synthesis, we adopt volume rendering techniques to synthesize color image $\hat{I}$ and depth map $\hat{D}$. To be specific, we sample a set of points for each emitted camera ray in a coarse-to-fine manner [15] and accumulate the radiance and the distance along the corresponding ray to calculate the rendered color $\hat{I}$ and depth $\hat{D}$. To obtain the radiance of a spatial point $\mathbf{x}_{\text{pts}}$, we use the nearest scene model for prediction. A set of per-image appearance embedding [13] is also optimized simultaneously in the training.

$$\hat{I}(\mathbf{o}, \mathbf{d}) = \int_{\text{near}}^{\text{far}} T(t)\sigma^{(k)}(\mathbf{x}(t)) \cdot c^{(k)}(\mathbf{x}(t), \mathbf{d}) \mathrm{d}t, \ \hat{D}(\mathbf{o}, \mathbf{d}) = \int_{\text{near}}^{\text{far}} T(t)\sigma^{(k)}(\mathbf{x}(t)) \cdot t \mathrm{d}t, \tag{7}$$

where $\mathbf{o}$ and $\mathbf{d}$ denote the position and orientation of the sampled ray, $\mathbf{x}(t) = \mathbf{o} + t\mathbf{d}$ represents the sampled point coordinates in the world space, and $T(t) = \exp\left(-\int_{\text{near}}^{t} \sigma^{(k)}(\mathbf{x}(s))\mathrm{d}s\right)$ is the

---

[4]Manual selection of weights requires laborious tuning, but comparable performance can be achieved.

[5]Note that the average velocity refers to the mean value calculated from the ground-truth camera pose of the current frame and the two adjacent frames, rather than the average value in the whole sequence.

accumulated transmittance. We optimize the scene representation model with only the photometric error as $\mathcal{L}_{\text{bootstrarp}} = \text{MSE}(I, \hat{I})$. We empirically observe that this bootstrapping is critical to the challenging third stage which jointly learns $\theta$ and $\phi$ using asynchronous RGB-D data.

### 4.3 Joint Optimization

While the time-pose function learns a good initialization from the RGB sequence, there are still errors to be compensated. In this section, we describe how we perform simultaneous mapping and pose optimization, which compensates for the initial error of the time-pose function.

We jointly optimize the inaccurate camera poses and the implicit maps: when fitting parameters $\Theta_{\text{NeRF}}^{(k)}$ of the scene representation, the estimated depth camera poses $\hat{\mathcal{T}}_D^{(j)} \in \text{SE}(3)$ (where $t \in \mathbb{R}^3$ and $q \in \text{SO}(3)$) will be simultaneously optimized on the manifold:

$$\theta, \{\hat{\mathcal{T}}_D\} = \underset{\theta, \mathcal{T} \in \text{SE}(3)}{\arg\min} \mathcal{L}(\{I^{(i)}\}, \{D^{(j)}\} \mid \theta, \{\hat{\mathcal{T}}_D\}), \tag{8}$$

where $\mathcal{L}$ is the objective function we demonstrate in the next paragraph.

To train the implicit representation to obtain photo-realistic RGB rendering maps and accurate depth map estimation, we update the mapping losses as:

$$\mathcal{L} = \lambda_{\text{color}} \sum_i \text{MSE}(I^{(i)}, \hat{I}^{(i)}) + \lambda_{\text{depth}}(\alpha) \sum_j \text{MSE}(D^{(j)}, \hat{D}^{(j)}), \tag{9}$$

where $\lambda_{\text{color}}$ and $\lambda_{\text{depth}}(\alpha)$ are weighting hyper-parameters for color and depth loss, in which the depth loss weight starts to grow from zero gradually with the training process $\alpha$.

To compensate for the error from the time-pose function extracted poses, we jointly optimize two implicit representation networks thanks to the end-to-end differentiable nature.

## 5 Asynchronous Urban Scene (AUS) Dataset

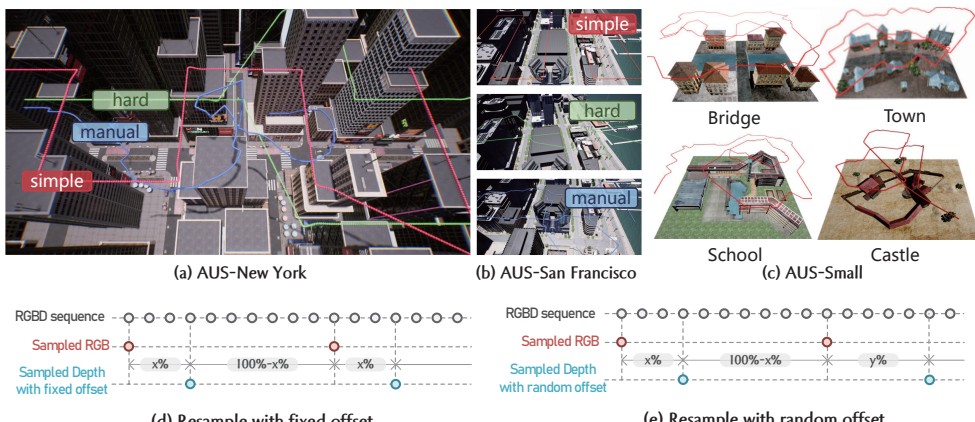

Figure 4: We propose a photo-realistically rendered dataset named Asynchronous Urban Scene (AUS) for evaluation. (a/b) are large-scale city scenes designed according to New York and San Francisco while (c) is (relatively) small-scale scenes provided by UrbanScene3D. Drone trajectories of different difficulty levels are visualized in (a-c). On these trajectories, we first capture an RGB-D sequence with an enough high framerate. Then we exploit two resampling strategies: fixed offset (d) and random offset (e). $x$ equals 30 in (d) for every RGB-D pair. $x$ equals 30 while $y$ equals 50 in (e).

**Dataset Collection.** Our Asynchronous Urban Scene (AUS) dataset as illustrated in Fig.4 is generated using Airsim [23], a simulator plug-in for Unreal Engine. With 3D city models loaded in Unreal Engine, the simulator can output photorealistic and high-resolution RGB images with synchronized depth images (resampled later) according to the a drone trajectory and a capture framerate. We choose Airsim as it strikes a good balance between rendering quality and dynamics modeling flexibility.

| Scene | Method | PSNR ↑ | SSIM ↑ | LPIPS ↓ | RMSE ↓ | RMSE log ↓ | $\delta_1(\%)$ ↑ | $\delta_2(\%)$ ↑ | $\delta_3(\%)$ ↑ |
|---|---|---|---|---|---|---|---|---|---|
| **NY** Mean | NeRF-W | 23.32 | 0.8105 | 0.2249 | 17.40 | 0.2630 | 80.17 | 90.11 | 94.72 |
| | Mega-NeRF | 23.53 | **0.8375** | 0.1920 | 23.99 | 0.2943 | 80.11 | 88.77 | 92.87 |
| | Ours | **24.33** | 0.8346 | **0.1833** | **6.15** | **0.0816** | **94.85** | **98.23** | **99.22** |
| **SF** Mean | NeRF-W | 19.21 | 0.6610 | 0.3632 | 24.93 | 0.1877 | 81.81 | 91.54 | 96.93 |
| | Mega-NeRF | 20.53 | 0.7334 | **0.2619** | 23.56 | 0.1713 | 88.58 | 94.74 | 96.83 |
| | Ours | **22.14** | **0.7930** | 0.2620 | **7.64** | **0.0789** | **96.34** | **98.80** | **99.70** |
| **Bridge** | NeRF-W | 26.79 | 0.8053 | 0.2438 | 131.88 | 1.2277 | 53.76 | 61.57 | 58.98 |
| | Mega-NeRF | 27.98 | 0.8674 | **0.1548** | 120.41 | 1.3246 | 69.10 | 72.54 | 73.17 |
| | Ours | **29.06** | **0.8751** | 0.1952 | **26.56** | **0.3248** | **93.24** | **96.32** | **98.26** |
| **Town** | NeRF-W | 21.32 | 0.6208 | 0.4088 | 132.70 | 1.4640 | 44.89 | 55.68 | 57.90 |
| | Mega-NeRF | 24.69 | 0.7305 | 0.3103 | 129.50 | 1.4240 | 54.54 | 59.18 | 57.90 |
| | Ours | **25.32** | **0.7675** | **0.2631** | **15.61** | **0.4632** | **91.92** | **96.89** | **98.49** |
| **School** | NeRF-W | 19.69 | 0.5715 | 0.4453 | 88.83 | 0.9365 | 61.73 | 72.58 | 75.80 |
| | Mega-NeRF | 25.57 | 0.7739 | 0.3191 | 63.10 | 0.7651 | 77.18 | 85.02 | 86.69 |
| | Ours | **26.51** | **0.7971** | **0.3175** | **21.19** | **0.2083** | **92.87** | **95.78** | **97.51** |
| **Castle** | NeRF-W | 22.63 | 0.7443 | 0.2557 | 78.18 | 0.8651 | 75.72 | 79.26 | 81.11 |
| | Mega-NeRF | 28.06 | **0.9053** | 0.1159 | 54.99 | 0.6167 | 79.69 | 83.59 | 87.43 |
| | Ours | **28.21** | 0.8976 | **0.1113** | **16.66** | **0.3565** | **93.12** | **97.23** | **98.45** |
| **Mean** | NeRF-W | 21.80 | 0.7156 | 0.3118 | 55.86 | 0.5846 | 72.20 | 81.40 | 84.88 |
| | Mega-NeRF | 23.85 | 0.7990 | 0.2262 | 51.06 | 0.5527 | 78.66 | 85.09 | 87.43 |
| | Ours | **24.85** | **0.8220** | **0.2223** | **12.14** | **0.1834** | **94.47** | **97.73** | **98.95** |
| | | (+1.00) | (+0.0230) | (-0.0039) | (-38.92) | (-0.3693) | (+15.11) | (+12.64) | (+11.52) |

Table 1: We quantitatively evaluate our method on the AUS dataset. Our method can synthesize more realistic images and more accurate depth maps than the baseline methods. For the **NY** and **SF**, we only report the mean performances on all sequences (Simple / Hard / Manual) due to limited space and more detailed results are in the supplementary materials.

**3D City Scene Models.** To generate the AUS dataset, we exploit a total of six scene models, covering two large-scale ones shown in Fig. 4-a/b and four (relatively) small-scale ones shown in Fig. 4-c. The former uses the New York and San Francisco city scenes provided by Kirill Sibiriakov [24], in which AUS-NewYork covers a $250 \times 150 m^2$ area with many detailed buildings and AUS-SanFrancisco consists of a $500 \times 250 m^2$ area near the Golden Gate Bridge. The latter uses four model files provided in the UrbanScene3D dataset [12]. As such, at the scene level, AUS features a good coverage of both large-scale modern cities and smaller cultural heritage sites.

**Trajectory Generation.** Trajectory complexity matters for our problem. On one hand, in many real-world applications, photographers may manually control drones to capture a city. On the other hand, simple trajectories can be modeled by simple functions, rendering the neural time-pose function unnecessary. To build a meaningful and comprehensive benchmark, we use three types of trajectories: a trivial Zig-Zag trajectory (simple in Fig. 4), a more complex randomly generated trajectory (hard in Fig. 4), and a very complex manually controlled trajectory (manual in Fig. 4). In AUS-Small, we only provide manually controlled trajectories, since the scene sizes are relatively small and using the former two trajectory strategies leads to an unrealistically large overlap between frames.

**Mismatch Resampling.** We first sample synchronized RGB-D sequences in the simulator at a high frequency (50fps) then re-sample RGB and depth images with various offsets to create asynchronous RGB-D sequences. As shown in Fig. 4-d/e, we exploit two settings for the AUS dataset. In Fig. 4-d, every RGB-D pair is resampled according to a fixed offset denoted by x%. For example, we sample the RGB image at 5fps or say every 10 frames and x = 30 means every depth image is 3 frames later than the RGB counterpart. In Fig. 4-e, the offset between an RGB-D pair is randomly selected, simulating a challenging real-world asynchronous sequence. Offset ablation will be shown later.

# 6  Experiments

In this section, we show the effectiveness of our 3-step optimization pipeline by qualitatively and quantitatively evaluating our proposed methods and comparing with baseline methods.

## 6.1  Results

We evaluate our proposed method against NeRF-W [13] and city-scale Mega-NeRF [29] and present the quantitative results in Table 1 and Table 2. NeRF-W is the baseline from which we borrow the aforementioned idea of per-image appearance embedding and Mega-NeRF is a state-of-the-art (SOTA) large-scale scene modeling framework which our network is built upon.

| Scene | Time-Pose Function | | Joint Optimzation | |
|---|---|---|---|---|
| | Rot. (°) | Trans. ($m$) | Rot. (°) | Trans. ($m$) |
| **NY Full** | 0.66 / 0.59 / 3.70 | 1.84 / 1.12 / 0.46 | 0.13 / 0.09 / 1.47 | 0.34 / 0.56 / 0.20 |
| **SF Full** | 0.17 / 0.67 / 0.65 | 1.34 / 1.45 / 0.94 | 0.05 / 0.41 / 0.02 | 0.32 / 1.09 / 0.66 |
| **Small** | 1.51 / 0.68 / 0.70 / 1.05 | 0.95 / 1.35 / 0.89 / 0.38 | 0.49 / 0.36 / 0.68 / 0.38 | 0.57 / 0.85 / 0.56 / 0.12 |
| **Mean** | **1.04** | **1.07** | **0.41 (-0.63)** | **0.53 (-0.54)** |

Table 2: We show that the time-pose function learns an accurate implicit trajectory from the RGB sequence that can estimate accurate poses for depth frames. By further tuning the time-pose function jointly with the scene representation network, the accuracy of the predicted depth sensor poses can be improved. The results of Simple / Hard / Manual on **NY** and **SF** are shown in the first two lines. The results of the 4 small scenes of (Bridge / Town / School / Castle) are shown at the bottom.

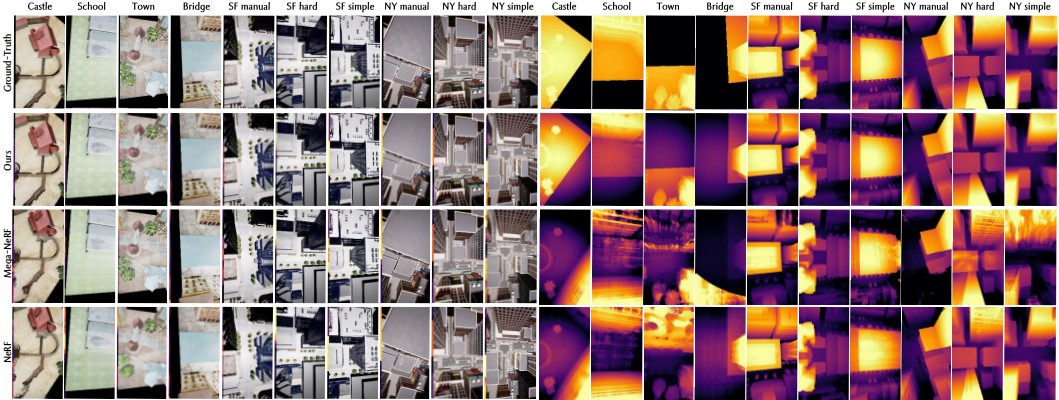

Figure 5: **Qualitative Results.** Our method can render photo-realistic novel views and the best depth estimation results. Please zoom in to see details using an electornic verion.

**RGB-D View Synthesis** The standard metrics for novel view synthesis and depth estimation are used for evaluation. For RGB view synthesis, metrics including PSNR, SSIM, and the VGG implementation of LPIPS [40] are used. For depth estimation, RMSE, RMSE log, $\delta_{1,2,3}$ are used. In Table 1, we quantitatively show on the RGB view synthesis task that, together with depth supervision, our method can generate more photo-realistic images than two SOTA baselines, and show on the depth estimation tasks that our method significantly improves the learned geometry of the scene representation network. We also present the RGB-D view synthesis results qualitatively in Fig.5 in which our method synthesize photo-realistic images and accurate depth maps, while baseline methods fail at predicting reliable depth maps (e.g. in the **School** scene, they mispredict the void space as a dense surface; in **NY hard**, the depth values around glasses are obviously inaccurate).

**Depth Pose Estimation** We evaluate the performance of our time-pose function in Table 2, or say specifically the accuracy of our method to localize depth sensor poses. As shown quantitatively, our method can achieve an average pose error of $1.04m$ and $1.07°$ in the first stage, where only time-pose pairs from the RGB sequence are used to optimize the network. After joint optimization in the third stage, our method cuts half the errors to $0.53m$ and $0.41°$.

**Real-world Evaluation.** In the real-world experiments, we use the DJI M300 UAV (equipped with a high-definition RGB camera and LiDAR to collect real data, where the RGB camera collects images at the frame rate of 30fps and the LiDAR collects depth information at 240Hz. The poses of the RGB images are provided by COLMAP [22]. The fixed transformations between sensors are provided by the producer or can be calibrated manually. A qualitative comparison is provided in Fig. 1 and more results are in the supplementary.

## 6.2 Ablation Studies

**Time-Pose Function Network Structure.** We compared our proposed time-pose function implementation with several commonly used implicit representation architectures on the localization accuracy

| Method | Rotation (°) | | Translation ($m$) | |
|---|---|---|---|---|
| | Mean | Median | Mean | Median |
| MLP | 26.62 | 17.53 | 8.23 | 7.50 |
| Feature Grid | 15.86 | 14.56 | 8.53 | 7.48 |
| Ours (L=1) | 24.24 | 12.99 | 9.20 | 8.01 |
| Ours w/o speed | 11.96 | 11.21 | 19.95 | 12.28 |
| Ours | **11.36** | **11.17** | **6.29** | **4.03** |

Table 3: Ablation on different network structures and the use of speed optimization.

| Offset | Rotation (°) | | Translation ($m$) | |
|---|---|---|---|---|
| | Mean | Median | Mean | Median |
| 10% | **0.66** | **0.26** | **3.42** | **1.88** |
| 20% | 0.94 | 0.55 | 4.96 | 3.90 |
| 30% | 1.24 | 0.79 | 6.41 | 5.78 |
| 40% | 1.41 | 0.75 | 7.26 | 6.61 |
| 50% | 1.50 | 0.84 | 7.53 | 6.51 |
| Random | 1.12 | 0.52 | 5.17 | 4.05 |

Table 4: Results on the ablation of different sampling offset strategies.

(Tab. 3). In order to highlight the gap between different methods, we downsample the dataset to increase the difficulty. (a) **Pure MLP architecture** : processing the positional-encoded [15] input timestamps with an MLP; (b) **1-D Feature Grid**: storing a feature vector for each second in the timestamp span and performing linear feature interpolation in the query's neighborhood. (c) **Ours**: our proposed 1-D multi-resolution hash grid with different resolution layers. The results (Table 3) show that our proposed multi-resolution architecture outperforms other network architectures in accuracy. The network structures are further detailed in the supplementary materials.

**Speed Loss.** We compared our method's localization accuracy with and without the optimization of motion speed (see the comparison of 'Ours' and 'Ours w/o speed' in Tab. 3). The results show that minimizing the gradient error (i.e., speed loss) help a lot in improving the accuracy of the translation (from $20m$ to $6.3m$).

**Different Sampling Offsets.** To show the robustness of our proposed time-pose function, we compare the localization accuracy of implicit trajectory representations under different sampling offsets, whose definition is described in Fig. 4-d/e. The quantitative results (Table. 4) indicate that the network's output exhibits a controllable margin of error as the data offset increases.

| Scene | Ours | | Mega-NeRF | | Mega-NeRF-Depth | |
|---|---|---|---|---|---|---|
| | PSNR ↑ | RMSE ↓ | PSNR ↑ | RMSE ↓ | PSNR ↑ | RMSE ↓ |
| **NY** Mean | **24.24** | **5.93** | 24.03 | 42.15 | 19.70 | 15.94 |
| **SF** Mean | **22.70** | **7.26** | 20.00 | 32.17 | 19.07 | 11.39 |
| **Bridge** | **29.06** | **26.55** | 27.98 | 120.41 | 22.35 | 96.16 |
| **Town** | **25.32** | **15.61** | 24.69 | 129.50 | 20.14 | 81.99 |
| **School** | **26.51** | **21.19** | 25.57 | 63.10 | 21.91 | 42.74 |
| **Castle** | **28.22** | **16.66** | 28.06 | 54.99 | 23.23 | 38.90 |
| **Mean** | **26.01** | **15.53** | 25.01 | 73.72 | 21.07 | 47.85 |

Table 5: Ablation on the joint optimization stage. We show that jointly optimizing the time-pose function and the scene representation significantly helps reduce geometric error.

**Joint Optimization for Pose Error Compensation.** To demonstrate the importance of rectifying erroneous poses of depth images in asynchronous RGB-D sequences using the time-pose function, we train a Mega-NeRF[29] with depth supervision but disabled the joint optimization stage. From the evaluation results (Table 5), we observe its substantial impact on the rendering quality (PSNR for RGB and RMSE for depth). Due to limited space, qualitative results are in the supplementary.

# 7 Conclusion

In this paper, we present a method to learn depth-supervised neural radiance fields from asynchronous RGB-D sequences. We leverage an important prior that the sensors cover the same spatial-temporal footprints and propose to utilize this prior with an implicit time-pose function. With a 3-staged optimization pipeline, our method calibrates the RGB-D poses and trains a large-scale implicit scene representation. Our experiments on a newly proposed large-scale dataset show that our method can effectively register depth camera poses and learns the 3D scene representation for photo-realistic novel view synthesis and accurate depth estimations. **Broader impact and limitations:** Large-scale scene modelling can be used for potential military use, which the method is not intended for.

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
