# OpenReview forum: "Learning Depth-regularized Radiance Fields from Asynchronous RGB-D Sequences"
_NeurIPS.cc/2023/Conference — Submitted to NeurIPS 2023_

### Official Review · Reviewer_9aLD · 2023-07-01

**Soundness:** 4 excellent
**Presentation:** 4 excellent
**Contribution:** 4 excellent
**Rating:** 6
**Confidence:** 4

**Summary:**

This paper explained and analysed the issue of unsynchronous RGB and depth measurement under the UAV city modelling scenario, which makes it challenge to incorporate depth supervision for NeRF optimisation under such scenerio. The authors proposed a novel solution to it by modelling the continuous RGB camera trajectory as an implicit time-pose function. Under the prior knowledge that both RGB and depth are generated on the same trajectory, poses of depth images can be queried from the time-pose function. Based on this time-pose function, the authors further proposed a 3-stage optimisation pipeline to train the NeRF model with depth supervision. Qualitative and quantitative results show the proposed method achieves better results compared with previous methods that use RGB inputs only.

**Strengths:**

1. This paper studied a very interesting and important problem of synchronising RGB and depth measurements, especially under large scale UAV scenerios, which posed challenges to incorporating depth supervision for NeRF optimisation under such scenerios.

2. The authors proposed a novel solution by modelling the continuous trajectory as a time-pose function, and designed a 3-stage optimisation pipeline to leverage the synchronised RGB and depth meaurement for NeRF training.

3. Experimental results showed the effectiveness of proposed methods compared to previous RGB-only methods. Ablation studies also show  naively using unsynchronised depth image could hinder the performance.

4. The paper is very well written and easy to follow. The storyline and motivation are very clear.

**Weaknesses:**

1. Although the experimental results in Tab. 1 and 5 showed the proposed method achieves better results than Mega-NeRF and NeRF-W, it's not compeltely fair as the baseline methods only take in RGB images. The authors need to show that the problem could not be easily solved by trivial efforts such as jointly optimising the poses of depth images. For example, in tab. 5 the author showed the results of Mega-NeRF with depth but the joint optimisation was switched off. Also in line 104, the authors mentioned that soft synchronization cannot fully address the misalignment issue, but it would be good to also show how good can all the methods perform with this simple alignment.

2. Most of the experiments and results are shown in synthetic dataset. Only Fig. 1 and the supplementary video showed results from real-world scenes. It would be good to show more results on real world sequences.

3. It would be good to show some run-time comparison and analysis.

**Questions:**

All my questions are stated in the Weakness section above.

**Limitations:**

Yes

---

> ### Author Rebuttal · Authors · 2023-08-09
>
> We thank 9aLD for recognition. The detailed responses to the weakness section are as follows:
>
> - **W1**: We present additional experiments and ablation studies in the author rebuttal section B, which demonstrates that our solution out-performs the baseline of directly using RGB poses as initialization for depth poses.
> - **W2**: We are sorry that we cannot provide more results on real-world sequences during rebuttal. Our code release provides interfaces that support experimentation on different data.
> - **W3**: We thank for this suggestion. Our time-pose function can be trained and evaluated in around 5 minutes (see also author rebuttal section A for runtime comparison), and the training of the neural radiance field may take days to complete.

---

> > ### Comment · Reviewer_9aLD · 2023-08-19
> >
> > I would like to thank the authors for their effort in providing the response. The authors have addressed most of my concerns for which I raise my ratings to Weak Accept.

---

### Official Review · Reviewer_JhZu · 2023-07-03

**Soundness:** 3 good
**Presentation:** 4 excellent
**Contribution:** 3 good
**Rating:** 7
**Confidence:** 4

**Summary:**

The paper proposes a way to train NeRF with asynchronous RGBD videos. Specifically, three technical contributions have been made:
1. New problem formulation for NeRF training from async RGBD video.
2. Propose a time-pose function to use async RGB and Depth stream, resulting in better pose estimation and NeRF training.
3. Propose a new synthetic dataset for this task.

---
**After rebuttal**: I have read authors' rebuttal and it addresses my concerns.

**Strengths:**

1. The paper is well-written and easy to follow.
2. It is novel to formulate the pose estimation issue of an RGBD stream in a time-pose function, which constrains the challenging issue better. Since ground truth poses for RGB stream is known, this paper essentially proposes a novel way to use an async depth stream to
    1. improve NeRF quality;
    2. refine RGB poses;
    3. optimise poses for depth images from interpolated RGB poses.
3. Although the method requires poses for the RGB video, which is a strong assumption for un-posed NeRF training, the formulation is still novel and smart to me.

**Weaknesses:**

In real datasets, I think there should be a self-motion distortion caused by the motion of the mounted drone. I am wondering how would this LiDAR depth distortion affect the performance?

**Questions:**

See the weakness section.

**Limitations:**

Yes.

---

> ### Author Rebuttal · Authors · 2023-08-09
>
> We thank JhZu for recognition and pointing out the depth distortion issue in real-world depth data. Addressing the depth distortion problem will be a promising future work but is not currently considered in this project. Our responses are three-fold:
>
> (1) During the capture of our real-world data, the drone moves in a slow enough speed so that the distortion is neglected.
>
> (2) According to the continuous time LiDAR SLAM literature like [A], explicitly modelling the distortion can improve RTE from 0.71% to 0.55% on KITTI-raw, which demonstrates that the impact of modelling LiDAR distortion (despite an important problem) is relatively small.
>
> (3) A recent study shows that improvement in depth maps' quality results in a marginal improvement in the quality of the reconstructed NeRF [B].
>
> [A] CT-ICP: Real-time elastic LiDAR odometry with loop closure, ICRA 2022
>
> [B] Digging into Depth Priors for Outdoor Neural Radiance Fields. ACM-MM 2023
>
> Summary: Despite an important problem to consider in the future, we think depth distortion does not substantially impact the scientific and practical value of this specific study.

---

> > ### Comment · Reviewer_JhZu · 2023-08-16
> > **Thanks for the rebuttal**
> >
> > Yeah, I started thinking of continuous SLAM literature while reading this paper. That's why I was asking about the effect of depth distortion. Nevertheless, the depth-distortion issue does not affect my rating. The problem setup is interesting to me and the results are supportive so I'll raise my ratings.

---

> > > ### Author Response · Authors · 2023-08-18
> > > **Thanks for the recognition**
> > >
> > > We thank JhZu for recognition and support.

---

### Official Review · Reviewer_GDqd · 2023-07-05

**Soundness:** 2 fair
**Presentation:** 2 fair
**Contribution:** 2 fair
**Rating:** 5
**Confidence:** 4

**Summary:**

The authors propose a method to reconstruct aerial scenes with Neural Radiance Fields that are supervised with RGB images and asynchronous depth images. To address the asynchronicity they propose a novel time-pose function that provides a prior to optimise the poses of the depth images. To validate their method a new synthetic dataset is introduced. They outperform one relevant baseline and show that asynchronous depth captures are an issue worth considering in NeRF reconstructions.

**Strengths:**

* S1. Relevancy. Depth supervision is an important method to improve 3D NeRF reconstruction and asynchronous depth has not been considered before. The Problem formulation is novel and interesting with significance in the field of UAV reconstruction.
* S2. Method Idea. The proposed time-pose function is an original approach to exploit the prior that depth and rgb captures lie on the same trajectory and it is shown that jointly optimising the NeRF reconstruction and time-pose function can significantly improve RGB and Depth reconstruction.
* S3. Presentation. The paper is written in a cohesive manner with a clear structure.

**Weaknesses:**

* W1. Missing baselines for pose optimisation. The authors ablate design decisions for their proposed time-pose function but do not provide baselines for pose optimisation, which is, in this reviewers opinion, their main contribution. A qualitative comparison to methods like BARF (https://github.com/chenhsuanlin/bundle-adjusting-NeRF) or other state-of-the-art methods (https://nope-nerf.active.vision/, https://prunetruong.com/sparf.github.io/) would support the decision to use the time-pose function.
* W2. Questionable assumptions for poses. As far as this reviewer understands the poses for the depth are assumed unknown and only initialised by the time-pose function that was trained on the rgb capture timesteps. This reviewer questions this assumption in general, as in both the synthetic and real-world setting poses for the depth can either be obtained from the simulation or from GPS measurements. A quantitative comparison between GT/GPS poses, BARF optimised poses and the time-pose function would strongly support the authors decision to use an implicit function to represent the drones trajectory.
* W3. Missing Implementation details for reproducibility. Some important implementation details are missing from both the paper and supplementary, to be precise:
   * The number of images & depth images in the generated and real datasets.
   * Is the time-pose function using positional encoding / fourier features ? This is not clear.
   * No training parameters are given for the proposed method and baselines.
* W4. Missing Discussion of related work. Related work in the field of representing signals with MLPs is not discussed (e.g. SIREN) and not considered in the design of the time-pose function.

**Questions:**

* Q1. How many images and depth images are used for training ?
* Q2. Why is a comparison to BARF, related methods and a GPS baseline not necessary for the proposed method - or why has it been omitted ?
* Q3. What is the intuition for the fully MLP-based baseline for the time-pose function ? The model capacity seems very large to represent a rather simple data distribution.
* Q4. Is the NeRF example in Figure 5 the NeRF-W baseline mentioned in the quantitative results or the standard NeRF ? This might be a typo but makes that part unclear.

UPDATE: After the rebuttal most of these questions have been addressed (see discussion for details). The score is adjusted and the authors are asked to include these additional clarifications into the final paper version.

**Limitations:**

The authors do not discuss limitations of their method in the paper but include a very small discussion in the supplementary. Overall limitations cannot really be discussed since there is no objective baseline to compare the method to. For broader impact only potential military uses are mentioned, whereas the method also has implications for surveillance. Environmental impacts of large-scale neural network training are also not mentioned.

---

> ### Author Rebuttal · Authors · 2023-08-04
>
> - **W1**: We provide additional experiment results that use the misaligned RGB poses (with calibrated sensor transformation $\mathcal{T}_{RGB\rightarrow D}$) as the initial value of depth sensor poses and perform BARF-style joint optimization. The quantitative results can be found in the Author Rebuttal section B above, which shows that this baseline method underperforms our pipeline.
> - **W2 & Q2**: Our experiment platform is the DJI-M300 UAV. The RGB pose information directly comes within the image EXIF data, while the depth pose cannot be directly obtained. This is because: 1. the RTK is triggered along with the RGB camera, but is misaligned with the depth sensor trigger; 2. the data collection software is close-sourced.
> - **W3 & Q1**: We thank GDqd for these feedbacks. (Q1 & W3-1) We present the number of RGB/D images of each sequence in the table at the bottom. As for positional encoding/the Fourier features, we would like to recap that we use a 1D hash-grid-based architecture for the time-pose function, which involves a small MLP head. The 1D hash-grid is locality-sensitive so that additional positional encoding is not necessary and its multi-scale nature naturally covers the role of Fourier features.  (W3-3) Some of the implementation details are listed at the bottom. More details can be found in the scripts in our open-sourced codebase.
> - **W4 & Q3**: We will augment our related works in the updated version, covering SIREN and other MLP-based implicit representation architectures. We also provide an experiment result of using SIREN to implement the time-pose function in the Author Rebuttal section A. It shows that our time-pose function (1d hash-grid based) out-performs SIREN.
> Finally, we would like to clarify that our implementation of the time-pose function is not MLP-based, but rather a 1-D hash grid, which is efficient in memory consumption. We choose a fully MLP-based baseline as it is a natural choice.
> - **Q4**: The method used in Fig.5 was NeRF-W. Thank you for correcting this. We will fix this in the camera-ready version.
> - **Limitation**: Thank you for pointing this out! We will supplement this part in our final paper.
>
>
> |               |        | \# of RGB Images | \# of Depth Maps | \# of Downsampled Frames | \# of Training Frames | \# of Evaluation Frames |
> | ------------- | ------ | ---------------- | ---------------- | ------------------------ | --------------------- | ----------------------- |
> | New York      | Simple | 2816             | 2816             | 57                       | 45                    | 12                      |
> |               | Hard   | 4176             | 4176             | 84                       | 73                    | 11                      |
> |               | Manual | 5532             | 5532             | 111                      | 100                   | 11                      |
> | San Francisco | Simple | 8071             | 8071             | 162                      | 151                   | 11                      |
> |               | Hard   | 20068            | 20068            | 112                      | 101                   | 11                      |
> |               | Manual | 13636            | 13636            | 273                      | 262                   | 11                      |
> | Real          |        | 284              | 284              | 284                      | 273                   | 11                      |
> | Castle        |        | 3633             | 3633             | 73                       | 62                    | 11                      |
> | School        |        | 8339             | 8339             | 167                      | 156                   | 11                      |
> | Town          |        | 5908             | 5908             | 119                      | 108                   | 11                      |
> | Bridge        |        | 11196            | 11196            | 224                      | 213                   | 11                      |
> | Mean          |        | 7605             | 7605             | 151                      | 140                   | 11                      |
>
> - **Implementation Details**: (Time-Pose Function) In our experiments, we maintain a multi-resolution feature grid and a shallow MLP decoder to form the time-pose function. We optimize the network via stochastic gradient descent, with an initial learning rate of $5\times 10^{-4}$ decaying exponentially to $5\times 10^{-5}$. (Joint-Optimization) We train all models for 500K iterations and render a batch of 1024 rays at each step, with a learning rate of $5\times 10^{-4}$ decaying to $5\times 10^{-5}$ for the scene representation networks, and $1\times 10^{-6}$ decaying to $1\times 10^{-7}$ for the pose optimization module.

---

> > ### Comment · Reviewer_GDqd · 2023-08-16
> >
> > Thanks for the clarifications. This indeed answers most of the questions I had.
> > I would encourage the authors to include the additional implementation details and data overview in the paper.
> >
> > Please find in the common section above the remaining questions on W1 as well as W4/Q3 as questions/comments to your experiments.

---

> > > ### Author Response · Authors · 2023-08-18
> > >
> > > We thank GDqd for the feedback. As for the concerns raised in the common section, please refer to the official comments Explanation of the inferior performance of SIREN & the BARF-style joint optimization.

---

### Official Review · Reviewer_2eof · 2023-07-05

**Soundness:** 3 good
**Presentation:** 3 good
**Contribution:** 3 good
**Rating:** 6
**Confidence:** 4

**Summary:**

This paper proposes to include depth supervision in NeRF in the UAV city modeling scenario. The key problem is that the images and depth maps are asynchronous. This paper exploits a prior that RGB-D frames are sampled from the same physical trajectory. It fits a time-pose function to the available RGB cameras and computes depth map cameras by this function and a pre-calibrated pose transform between sensors. Then it trains a NeRF with RGB loss and optimizes it further with RGB-D supervision. A new synthetic dataset is proposed for evaluation.

**Strengths:**

1. This paper tackles a new problem in NeRF modeling on UAV datasets. it defines a scenario where RGB and depth frames are asynchronous.

2. Instead of directly adding new parameters to predict depth camera poses together with the NeRF optimization, it identifies a prior of the relation of RGB poses and depth poses.

3. It designs a network with 1D hash encoding to fit a time-pose function for RGB cameras.

4. It generates a new synthetic dataset to evaluate the proposed method.

5. It designs a new speed loss in pose learning.

**Weaknesses:**

1. A key contribution of this method is the learned time-pose function. It actually can be seen as an interpolation function to interpolate the RGB poses on depth timestamps. Since camera interpolation is a common practice in 3D software such as Blender, I am curious about whether other simple interpolation methods such as linear interpolation or the interpolation in Blender could get worse or better accuracy.

2. As stated above, on easier datasets, the simple interpolation methods may still get good results. It will be better to construct more challenging datasets to demonstrate the effectiveness of the proposed time-pose function. This situation can be considered when generating synthetic datasets.

3. Directly adding extra parameters to estimate depth poses are also an option as stated in the paper. What is the accuracy like if we directly use this method? It would be not surprising if it gets worse results because the poses are not well initialized.

4. The prior that "RGB-D frames are actually sampled from the same physical trajectory" can be explained more in the main paper such as in the Introduction section. This can make us understand clearly that there is a fixed and known pose transformation between the depth and image sensors so we can estimate the depth poses by interpolating image poses.

**Questions:**

1. What is the performance using simple linear interpolation or the Blender to get depth poses?

2. If the simple interpolation methods can still get good results, can we consider this and generate more challenging datasets to validate the effectiveness of the proposed time-pose function?

3. What is the performance if we directly add extra parameters to estimate the depth pose?

**Limitations:**

The authors addressed the limitations.

---

> ### Author Rebuttal · Authors · 2023-08-04
>
> We thank 2eof for recognition and professional feedbacks. Here are detailed responses:
>
> - **Weakness #1 & #2**: We provide another ablation experiment in the Author Rebuttal section A. As demonstrated by the experiment results, our time-pose function out-performs linear interpolation.
> - **Weakness #3**: Does 2eof mean "use the misaligned RGB poses (with calibrated sensor transformation) as the initial value of depth sensor poses and perform BARF-style joint optimization."? If it is, we provide the experiment results in the Author Rebuttal section B and show that directly refining depth poses from misaligned RGB pose get worse averaged results.
> - **Weakness #4**: We will refine our manuscript in the next version. We do agree that stating RGB-D sequences are from the same physical trajectory without highlighting the existence of a known transformation between two cameras can be confusing at the first glance. Thanks for this suggestion.

---

> > ### Comment · Reviewer_2eof · 2023-08-18
> > **Thanks for the rebuttal**
> >
> > I have read the rebuttal related to the questions of mine and other reviewers. The rebuttal has provided proper experiments and analysis to the raised questions and I think this paper is interesting. I keep my original score.

---

### Official Review · Reviewer_ScBX · 2023-07-08

**Soundness:** 1 poor
**Presentation:** 2 fair
**Contribution:** 2 fair
**Rating:** 3
**Confidence:** 4

**Summary:**

This paper proposes a pipeline to build depth-supervised neural radiance fields (NeRF) using asynchronous RGB-D sequences. Since the task is novel, the paper also contributes a synthetic dataset and demonstrates that it outperforms certain baselines in the experiments.


**Strengths:**

## Originality
The task is novel and practical for lots of real-world settings.


**Weaknesses:**

## Quality
- The most obvious baseline is missing. What if authors simply initialize the depth frames with their (misaligned) poses from the sensor and perform BARF [1] style joint optimization on the depth frames’ poses (note that depth frames can have different poses from the RGB images after optimization)?


- Since the above baseline is missing, it’s hard to understand why the time-pose network is necessary.

- In my opinion, the baselines authors compared against are in the wrong direction. NeRF-W and Mega-NeRF are not solving the problem of inaccurate camera poses. Authors should look into works that try to solve pose estimation and neural radiance fields simultaneously such as BARF, NeRF–, and iNeRF [1, 2, 3].

- Line 156-157’s motivation for using Quaternion since other rotation representations are not continuous is weird. Quaternion is also not continuous and is known to not be the best representation for rotation regression. See this paper [4]


### References
- [1] BARF: Bundle-Adjusting Neural Radiance Fields, Lin et al.
- [2] NeRF--: Neural Radiance Fields Without Known Camera Parameters, Wang et al.
- [3] INeRF: Inverting Neural Radiance Fields for Pose Estimation, Yen-Chen et al.
- [4] On the Continuity of Rotation Representations in Neural Networks, Zhou et al.


**Questions:**

Same as Weaknesses.

**Limitations:**

Yes

---

> ### Author Rebuttal · Authors · 2023-08-04
>
> ## Weakness 1
> > The most obvious baseline is missing. What if authors simply initialize the depth frames with their (misaligned) poses from the sensor and perform BARF [1] style joint optimization on the depth frames’ poses (note that depth frames can have different poses from the RGB images after optimization)?
>
> > Since the above baseline is missing, it’s hard to understand why the time-pose network is necessary.
>
> We provide additional experiment results that use the misaligned RGB poses (with calibrated sensor transformation $\mathcal{T}_{RGB\rightarrow D}$) as the initial value of depth sensor poses and perform BARF-style joint optimization. (Please see the Author Rebuttal section B above).
>
> ## Weakness 2
> > In my opinion, the baselines authors compared against are in the wrong direction. NeRF-W and Mega-NeRF are not solving the problem of inaccurate camera poses. Authors should look into works that try to solve pose estimation and neural radiance fields simultaneously such as BARF, NeRF–, and iNeRF [1, 2, 3].
>
> These methods calibrate the camera pose of RGB images, which is different from our setting, where poses of RGB images are known, but depth poses are unknown. Therefore, these methods cannot be readily applied in our setting as baseline methods. Moreover, our pipeline integrates the BARF method [1] in our joint optimization stage, and we have shown in Weakness #1 that directly using BARF-like joint optimization under-performs our pipeline.
>
> ## Weakness 3
> > Line 156-157’s motivation for using Quaternion since other rotation representations are not continuous is weird. Quaternion is also not continuous and is known to not be the best representation for rotation regression. See this paper [4]
>
> We will rephrase this sentence in an updated version, by weakening the claim of 'continuous' to 'we use quaternions as the representation because they have well-behaving training dynamics when compared with Euler angles/rotation matrices and are widely used in computer vision [A/B/C].
>
> We will cite [1-4] and explore the rotation representation in [4] in the future and hope this rephrasing could address the concern. Thank you very much!
>
> [A] Kendall, A., & Cipolla, R. (2017). Geometric Loss Functions for Camera Pose Regression with Deep Learning. 2017 IEEE Conference on Computer Vision and Pattern Recognition (CVPR), 6555–6564. https://doi.org/10/gg5v2h
>
> [B] Kendall, A., Grimes, M., & Cipolla, R. (2015). PoseNet: A Convolutional Network for Real-Time 6-DOF Camera Relocalization. 2015 IEEE International Conference on Computer Vision (ICCV), 2938–2946. https://doi.org/10/gc4n9z
>
> [C] Pavllo, D., Tan, D. J., Rakotosaona, M.-J., & Tombari, F. (2023). Shape, Pose, and Appearance From a Single Image via Bootstrapped Radiance Field Inversion. 4391–4401.

---

### Author Rebuttal · Authors · 2023-08-09

We thank all the reviewers for their professional and in-depth comments. We respond to two shared concerns here.

## A. Comparisons with linear interpolation and non-hash implicit network architectures (along with wall-clock runtime)

> The weakness 1&2 of reviewer 2eof, weakness 4 of reviewer GDqd, and weakness 3 of reviewer 9aLD.

We provide additional experiment results to evaluate alternatives to the proposed hashgrid-based time-pose function. The table below shows the quantitative result of our method compared to linear interpolation and a typical MLP-based implicit function approximator named SIREN [A]. The averaged rotation and translation errors and a rough runtime are reported.

[A] Implicit neural representations with periodic activation functions, NeurIPS 2020

|                         |Rotation($^\\circ$)|Translation(m)|Time     |Acceptable   |
|-------------------------|-------------------|--------------|---------|-------------|
|Linear Interp.           |10.1737            |1.7298        |< 10 sec.|$\\times$|
|SIREN                    |5.705              |144.3         |~10 min. |$\\times$|
|Time-Pose Function (ours)|0.3745             |0.0712        |~5 min.  |$\\checkmark$|

Specifically speaking, we directly perform linear interpolation on the translation vector and the Euler angle. The interpolated Euler angles are then converted to quaternions. We show that our time-pose function significantly out-performs these two alternatives. The linear interpolation baseline can achieve acceptable performances in estimating the translation vectors, but the estimated rotation is not as good. Empirically, a rotation error that is larger than 5$^{\circ}$ in UAV captured images is not acceptable as the initial value for the downstream joint optimization algorithm.

As for the comparison with SIREN, we use the official Python package to implement the time-pose function. The input time-stamp and output translation vectors are normalized in [-1, 1] range. We exhausively tune the hyper-parameters and report the best performance of different design choices (including network design, network capacity, etc.). It is shown that SIREN gives surprisingly inferior results on the task of time-pose function fitting.

Training our method for around 5 minutes with very low memory consumption achieves the best performance in estimating both rotation and translation.

## B. Directly using RGB poses as initialization for depth poses and performing joint optimization

We perform an ablation study and provide the quantitative results below which report averaged metrics. The results show that our method achieve better results by using the initial values generated by the time-pose function rather than directly using the transformed poses from the RGB sequence.

|                         |Rotation($^\\circ$)|Translation(m)|RGB PSNR |Depth RMSE |
|-------------------------|-------------------|--------------|-----|-----|
|RGB Init. w/ Joint Optim.|0.72               |1.378         |20.41|12.04|
|Ours full                |0.09               |0.56          |24.33|6.15 |

Intuitively, the RGB poses are good initial values when the drone motion is small (e.g. small rotation, slow motion, etc.) but perform worse when there's larger motion in the sequence (e.g. drone turning, flying at high speed, etc.).

---

> ### Comment · Reviewer_GDqd · 2023-08-16
> **Trying to understand the surprising (counterintuitive) results.**
>
> Thanks for providing this additional insights!
>
> The results, however, are both very surprising - and it would be helpful to understand better why:
> - A: Why is a SIREN encoding providing such a problematic translation error as it intuitively provides the same information as in "linear interpolation" but with finer granularity
> - B: Counterintuitively, you state that for small motions RGB poses are more helpful. Do you observe this in practice? Is there a pattern on when it over / underperforms the presented method? You mention in your answer to my review that you perform a "BARF-style joint optimization" which is a full BA-optimization. What is the reason for this being inferior to your proposal?
>
> Do you have explanations to these points? Can you comment on that?
>
> Thanks!

---

> > ### Author Response · Authors · 2023-08-18
> > **Explanation of the inferior performance of SIREN & the BARF-style joint optimization.**
> >
> > ### A
> > **Firstly, we would like to respond to the original comment**: ‘Related work in the field of representing signals with MLPs is not discussed (e.g. SIREN) and not considered in the design of the time-pose function.’
> >
> > We assume (with all due respect) that this implies: SIREN may perform better than a pure MLP network in modeling time-pose function due to its superior performance in modeling 3D data when compared with a pure MLP network.
> >
> > We would like to note that the reason why SIREN performs better than a pure MLP in representing 3D data is not only attributed to the sine-activated architecture but also: (1) the Eikonal constraint; (2) the trick of pushing randomly sampled points to infinity; (3) the normal constraint. These three losses are not applicable to the time-pose function, so we do not think that the superiority of SIREN over MLP can readily translate to the time-pose function.
> >
> > **Secondly, we would like to respond to the new comment**: ‘It intuitively provides the same information as in "linear interpolation" but with finer granularity.’
> >
> > We would like to note that since we have many RGB anchor poses, here ‘linear interpolation’ is actually piece-wise linear interpolation. Intuitively, a 1D-hash grid-based network, which is also piece-wise between two grid points, is much more similar to ‘linear interpolation’ than MLP/SIREN, because MLP/SIREN models the whole target function using a single network. So, SIREN actually works with a coarser granularity.
> >
> > **Thirdly, SIREN has other intrinsic limitations**:
> >
> > We provide more details of the experiments with SIREN. According to SIREN’s requirement on data distribution, we first normalize the timestamps and the translation vectors into [-1, 1] range by centering and scaling. The scale factor for the translation vectors is set to 200 in our experiment. We then construct a SIREN network using their official Python package with 5 SIREN layers and 256 channels in each layer. The initial weights for each layer are set according to the SIREN’s supplementary materials since it is critical for performance. An extra output layer with an identity activation function is used to predict the pose. This setting achieves the best performance although other settings are also tested throughout the experiments, including changing the network width/depth, using positional encoding, using different learning rates, etc. Through analysis, we conclude that the poor performance of the SIREN-based time-pose function is credited to the following reason:
> >
> > Due to the nature of the sine activation function, SIREN requires that the input timestamps and the output poses are normalized. By scaling the input & output back up to the real-world quantities, the prediction error is also scaled up. However, our method is robust to scale. The table below shows the performances of our method with/without data preprocessing (including centering and scaling).
> >
> > |                 | Ours w/o preprocessing  | Ours w/ preprocessing | SIREN  |
> > | --------------- | ---------------------------------------------- | ---------------------------------------------- | ------------------------------- |
> > | Translation (m) | 0.1821                                         | 0.0712                                         | 144.3                           |
> >
> >
> > ### B
> > We then respond to the new review that a "BARF-style joint optimization" which is a full BA-optimization:
> >
> > We assume (with all due respect) that this implies: (1) using a full BA optimization can wipe out the impact of pose initialization (using direct RGB initialization or time-pose function initialization); (2) using a BARF-style joint optimization (which a full BA-optimization) can wipe out the impact of pose initialization (using direct RGB initialization or time-pose function initialization).
> >
> > We have different opinions as such: (1) a traditional BA-optimization procedure, no matter it is full or not, is highly non-linear and non-convex. So a better pose initialization (e.g., generated by time-pose function) can lead to better BA results than a trivial pose initialization (e.g., generated by nearest RGB pose). (2) A BARF-style joint optimization, when considered as a full BA-optimization, introduces more non-linearity and non-convexity thanks to the nature of neural networks. So a BARF-style joint optimization also enjoys the benefit of a better pose initialization (e.g., generated by time-pose function).
> >
> > Finally, as a response to the question ‘Do you observe this in practice? Is there a pattern on when it over / underperforms the presented method?’, we would like to provide these facts: In real-world experiments, we find the RGB initialization works only when the drone is still (up to motion from vibration/wind)  when taking photos or flying slowly. A simple zigzag route (which is common for UAV data collection) causes the joint optimization stage to fail with only RGB initializations.

---

> > > ### Comment · Reviewer_GDqd · 2023-08-20
> > >
> > > Thank you for these additional insights!
> > > The scaling argument and the backup by practical experience sounds indeed convincing.
> > > I am very thankful for this extra effort of the authors and will increase my rating. Please include the additional experiments (especially regarding BARF etc.) and overview including a brief discussion of the potential implications with MLP-driven positional encoding in the final paper version.

---

### Comment · Area_Chair_JoRu · 2023-08-12
**Author-reviewer discussion starts**

Dear reviewers,

Thanks for serving as reviewers.

The authors have submitted a rebuttal. Please review through the rebuttal and reviews from other authors. If you have any questions, please feel free to let the authors know. You are more than welcome to post comments for further explanation or clarification before 1pm EDT on 8.21.

Best,

AC

---

### Comment · Area_Chair_JoRu · 2023-08-18
**Kind reminders for reviewers**

Dear reviewers,

If you have not responded to the authors' feedback, please take some time to read through their responses and reviews from other reviewers. We would be very pleased to hear your thoughts.

Thanks,

AC

---

### Decision · Program_Chairs · 2023-09-21

**Decision:**

Reject

**Comment:**

The submission received 4 positive recommendations and 1 negative recommendation. Initially, the reviewers were concerned about the evaluations, the missing details, and the performance on real-world data. The authors addressed most of the concerns in the rebuttal and the responses to reviewers' additional comments. The reviewers did not reach a consensus of acceptance after the discussion period. The AC read through the submission, the review, the rebuttal, and the discussions. The AC was also concerned about the comparisons with methods like BARF, Nerf– mentioned by reviewer ScBX. In the response to reviewer ScBX, the authors claimed that the method used for calibrating the camera pose of RGB images may not be able to calibrate depth images. This explanation does not make lots of sense to me. Since we can render both RGB and depth images using the same alpha blending in volumetric rendering, it should work if depth images are available for pose estimation in NeRF. Therefore, it would be great if the authors could report comparisons with the latest methods in a more fair way. Per this, the AC rejects this submission. The decision was discussed with and approved by the SAC. Hopefully, the reviewers’ comments and advice are helpful for the authors to improve the manuscript.